# Absorption Band Tunable La-Sr Co-Doped BaCo_2_-W Type Hexaferrites

**DOI:** 10.3390/ma16175897

**Published:** 2023-08-29

**Authors:** Juan Li, Xuetao Sun

**Affiliations:** 1College of Materials Science and Engineering, Zhejiang University of Technology, Hangzhou 310014, China; 2Research Center of Magnetic and Electronic Materials, Zhejiang University of Technology, Hangzhou 310014, China

**Keywords:** W-type hexagonal ferrite, microwave absorption materials, tunable absorption band, reflection loss

## Abstract

La-Sr co-doped Ba_1−x_(La_0.5_Sr_0.5_)_x_Co_2_Fe_16_O_27_ (x = 0.0, 0.2, 0.4, 0.6, 0.8 and 1.0, respectively) hexaferrites were prepared by the solid-state method. W-type hexaferrite single phase structure with space group P63/mmc was obtained when the doping amount was x < 0.4 and an M-type hexaferrite and a spinel phase with smaller grains gradually replaced the W phase as the primary phases when x ≥ 0.6. The maximum Ms is 76.2 emu/g and the minimum Hc is 60 Oe at x = 0.4, as obtained by VSM analysis. The magnetoelectric properties of the samples were tested at 1–18 GHz with a vector network analyzer and the reflection loss was calculated based on transmission line theory. It was found that the introduction of an appropriate amount of La-Sr provides a large number of porosity defects while increasing the grain size, which can effectively improve the reflection of electromagnetic waves inside the material and dissipate more energy. At the same time, co-doping also makes the resonance frequency of the samples shift to lower frequency, resulting in tunable absorption properties in the C, X and Ku bands. When x = 0.2, the minimum reflection loss is −40.61 dB at 1.5 mm thickness, with the effective absorption bandwidth of 5.76 GHz in the X band; when x = 0.4, the minimum reflection loss is −37.45 dB at 2.5 mm, with the bandwidth of 4.97 GHz in the C band; when x = 0.6, the material has good absorption in both the X and Ku bands with the thickness less than 2 mm. The simple preparation method and good performance make La-Sr co-doped Co_2_W ferrite a promising microwave absorbing material.

## 1. Introduction

While the development of information technology brings convenience to human life, the electromagnetic waves generated also have an impact on human health and the operation of electronic equipment [1,2,3,4]. In addition, the continuous development of radar detection technology puts forward higher requirements for the radar stealth effect of military equipment. Therefore, in recent years, more and more research has been done on microwave absorption and electromagnetic shielding materials. The ideal absorbing material has the characteristics of strong absorption, wide bandwidth and thin thickness [5,6]. Microwave absorbing materials generally include dielectric loss type and magnetic loss type. Dielectric loss is related to the leakage current or relaxation polarization process in an alternating electric field, while magnetic loss includes hysteresis loss, eddy current loss and residual loss. Compared with dielectric loss absorbing materials, magnetic loss absorbing materials usually have better absorbing properties due to the coexistence of dielectric loss and magnetic loss [7]. Ferrites are typical magnetic loss absorbing materials, which have the advantages of low preparation cost and strong adsorption and have been widely studied [8,9].

The W-type hexagonal ferrite, due to its lamellar structure, high magnetization strength, good planar magnetic crystal anisotropy and adjustable magnetic dielectric properties, is a promising microwave absorption material [10,11]. Co_2_-W type hexaferrites are superior to most of the spinel ferrites for their applications in microwaves and in different electromagnetic devices operating in the radio frequency region [3]. However, there are still many issues with the application of pure W-type ferrite, such as narrow effective bandwidth and thick thickness. Currently, the research on W-type hexagonal ferrite mainly includes ion doping and multiphase composites. Khan et al. synthesized BaCo_1.6_Mg_0.4_Fe_16_O_27_ by the chemical co-precipitation method, with a peak reflection loss of about −11 dB and a bandwidth of 1.73 GHz [12]. Ahmad et al. prepared Ba_0.5_Sr_0.5_Co_2-x_Me_x_Fe_16_O_27_ ferrite with different metal ion substitutions [13]. All samples had a hexagonal laminar structure, and their peak frequency of the reflection loss could be tuned by changing the thickness of the absorber. Liu et al. synthesized Sr-W ferrite and found that when a 2 mm coating composed of 20% nickel and 80% ferrite was applied, the minimum reflection loss at 12.08 GHz was −20.09 dB, with a bandwidth of 4 GHz [14].

As above, the microwave absorption performance of W-type ferrite can be improved through ion doping. Currently, most studies focus on single ion doping, while research on multi-ion co-doping is relatively limited. Jasbir et al. synthesized Co^2+^ and Al^3+^ substituted M-type hexagonal ferrites; they use a quarter wavelength mechanism and impedance matching mechanism to evaluate the absorbing effect [15]. Kaur et al. prepared cobalt- and indium-doped M-type barium–strontium hexagonal ferrites by the standard ceramic method. They found that wave absorption could be improved by adjusting the thickness and composition of ferrites [16]. Gui et al. prepared W-type Ba_1−x_La_x_Co_2_Fe_16−y_Al_y_O_27_ hexagonal ferrite by using the sol-gel auto combustion method and found that with increasing Fe^2+^ concentration and reduction of grain sizes, the dielectric loss of La–Al doped ferrite is larger than undoped Co_2_W hexagonal ferrite. In addition, the reflection loss of sample Ba_0.95_La_0.05_Co_2_Fe_15.5_Al_0.5_O_27_ reaches −50.13 dB at 11.9 GHz and 7.0 mm thickness [17]. Therefore, in the present work, Ba_1−x_(La_0.5_Sr_0.5_)_x_Co_2_Fe_16_O_27_ (x = 0, 0.2, 0.4, 0.6, 0.8 and 1.0, respectively) ferrite was prepared by the solid-phase method using La-Sr co-doping. The effects of La-Sr co-doping on phase composition, microstructure, magnetic properties and microwave performance were investigated. The results obtained demonstrate that La-Sr co-doped ferrite exhibits relatively good absorption performance.

## 2. Materials and Methods

Polycrystalline samples of W-type hexaferrites with general formula Ba_1−x_(La_0.5_Sr_0.5_)_x_Co_2_Fe_16_O_27_ (x = 0, 0.2, 0.4, 0.6, 0.8 and 1.0, respectively) were prepared by the solid-state reaction method. The high purity powder of BaCO_3_ (99%), SrCO_3_ (99.95%), Fe_2_O_3_ (99%), Co_3_O_4_ (99.9%) and La_2_O_3_ (99.99%) were weighed according to stoichiometric proportions and ball-milled with ZrO_2_ in a deionized water medium for 24 h to obtain a homogeneous mixture. The mixture was dried at 100 °C for 5 h under vacuum conditions and then calcined at 1270 °C for 2 h to obtain calcined powders with a primary phase of W-type hexaferrites. The calcined powders were then ball-milled for 16 h and dried. The powders were reground with 8 wt% polyvinyl alcohol (PVA) binder before being pressed uniaxially into toroidal specimens (outside diameter of 12.5 mm, inside diameter of 7.6 mm and thickness of 3–5 mm) and discs (diameter 10 mm), as shown in Figure 1. The samples were preheated at 550 °C for 30 min to expel the binder and then sintered at 1270 °C for 2 h at a heating rate of 5 °C/min.

X-ray diffraction analysis was carried out by a diffractometer (Rigaku Corporation, Tokyo, Japan) having CuKa as a radiation source. The surface morphology was analyzed by a Field-Emission Scanning Electron Microscope (FE-SEM, HITACHI Regulus 8100, Hitachi, Tokyo, Japan). Hysteresis loops for hexaferrite powders were recorded using a Vibrating Sample Magnetometer (VSM, LakeShore7404, Lakeshore, Westerville, OH, USA). The permeability and permittivity values were obtained from the S11 and S12 parameters measured via the coaxial line method in 1–18 GHz by a vector network analyzer (VNA, Agilent E5071C, Agilent, Santa Clara, CA, USA). For this measurement, the sintered ring-shaped specimens were machined to an outer diameter of 7 mm, an inner diameter of 3.04 mm and a thickness of 2 mm.

## 3. Results and Discussion

### 3.1. Phase Component

Figure 2 represents the XRD spectra of the sintered samples co-doped with LaSr (x = 0.0, 0.2, 0.4, 0.6, 0.8 and 1.0). By comparing with the standard cards, the phase compositions of each doped sample were determined. The main phase of the samples with doping amounts x = 0.0, 0.2, 0.4 and 0.6 was the W phase, consistent with the standard card 78–0135. The spinel phase and the M phase appear from x = 0.4 and increase with the increase in doping amount. As the doping amount increased to x = 0.8, the M phase and the spinel phase became the principle phases, with no W phase observed. The primary cause for the formation of the spinel and M phase is the charge imbalance brought about by replacing Ba^2+^ with La^3+^. To maintain balance, some Fe^3+^ transitions to Fe^2+^, and with the appearance of the M phase, CoFe_2_O_4_ is jointly generated by Co^2+^ and Fe^2+^. The phase formation temperature of SrW is higher than that of BaW, so after the doping amount x > 0.6, SrM becomes the main phase.

### 3.2. Microstructure

Figure 3 illustrates the cross-sectional morphology of samples with different doping amounts. There are no obvious pores in the grains in Figure 3a, while pores in the grains are present in Figure 3b–d and seem to increase with doping. The appearance of pores is maybe due to the fact that the ionic radii of La^3+^ (1.032 Å) and Sr^2+^ (1.18 Å) are smaller than that of Ba^2+^ (1.35 Å), which leads to defects due to lattice distortion during the growth of the grains and accelerates the grain growth rate. For absorbing materials, the presence of pores increases the reflection of electromagnetic waves inside the material, which is beneficial for attenuating electromagnetic waves. The grains in Figure 3e,f are all very small at around 5 μm with no obvious pores visible in the grains and the grain boundaries are not distinct. As shown in the XRD results above, the principle phases of samples with x = 0.8 and 1.0 become the M and spinel phase, which result in very different grain morphologies.

### 3.3. Magnetic Properties

Figure 4 shows the hysteresis loops of samples with different doping amounts, and the specific values of Ms and Hc are listed in Table 1. The saturation magnetization of pure Co_2_W hexaferrite is 73.1 emu/g, and Hc is 130 Oe. In the range of x = 0.0–0.4, as the doping amount increases, the saturation magnetization gradually increases, while the coercivity and remanent magnetization both decrease. When x = 0.4, the maximum Ms is 76.2 emu/g and the minimum Hc is 60 Oe. The magnitude of Ms mainly depends on the structure and composition of the material, and the type and occupancy of magnetic atoms in the lattice will affect the magnetic moment of the material, thus influencing the saturation magnetization. Replacing Ba^2+^ with smaller La^3+^ and Sr^2+^ will reduce the lattice constant, thereby reducing the distance of Fe-O parallel to the c-axis, strengthening the superexchange interaction and thus increasing Ms. In addition, the increase in internal stress caused by lattice distortion enhances the magnetic interaction in the sublattice, which also increases the magnetization strength [18]. The grain size affects the coercivity of the material. The smaller the grain, the greater the coercivity. From Figure 3a–c, it can be seen that the grains tend to grow larger, so the coercivity will gradually decrease. In addition, there is a certain relationship between coercivity and Ms, which can be expressed by Equation (1) [19].
(1)$$\mathrm{Hc}=\left(\frac{2K_{1}}{\mathrm{Ms}}\right)-\mathrm{NMs}$$
where K_1_, Ms, and N are the first anisotropy constant, saturation magnetization, and demagnetization factor, respectively. It can be seen that Hc is inversely proportional to Ms and decreases as Ms increases. When the doping amount x > 0.4, as the value of x increases, Ms gradually decreases, and Hc gradually increases, which is related to the appearance of the M phase. The M-type hexaferrite is a hard magnetic phase, which has a lower saturation magnetization and a higher coercivity. As mentioned above, as the doping amount increases, the M phase becomes the main phase, so the shape of the hysteresis loop will change towards the hard magnetic phase.

### 3.4. Microwave Properties

The complex permittivity (ε = ε′ − *i*ε″) and complex permeability (μ = μ′ − *i*μ″) are important parameters for measuring the performance of absorbing materials. The trend of dielectric changes with frequency in the range of 1–18 GHz for the samples is shown in Figure 5. With the increase of La-Sr doping amount, both ε′ and ε″ decrease first and then increase. Pores are one of the important factors affecting the dielectric constant. For samples with x = 0.4 and 0.6, high porosity in the grains results in a decrease in the dielectric constant. In addition, for samples with x = 0.8 and 1.0, the interface polarization is strengthened by the obvious reduction of grain size and the relaxation polarization causes the electron transition between Fe^3+^ and Fe^2+^, resulting in the increase of the dielectric constant and obvious resonance peaks. As shown in Figure 5c, the tangent of the dielectric loss angle, which can be used to represent the dielectric loss ability of the sample, is significantly higher for the samples with x = 0.8 and 1.0 than for other samples. This heightened dielectric loss in the sample is chiefly attributable to the considerable increase in electronic transitions between Fe^3+^ and Fe^2+^ within the spinel phase.

Figure 6a,b demonstrates the variation in magnetic permeability with frequency for samples with different doping amount. Grain size is one of the significant factors affecting the initial magnetic permeability of a material. The larger the grain size, the smaller the stress near the grain boundaries, leading to lesser impedance to the movement of domain walls, and subsequently, a higher initial magnetic permeability. From Figure 6a, it is observed that the initial magnetic permeability of the material increases first and then decreases, reaching its maximum at x = 0.4, which coincides with the variation pattern of the crystal size shown in Figure 3. As shown in Figure 6b, the resonance peaks of samples with x = 0.0–0.6 are located in the frequency range of 6–10 GHz, and the peak frequency shifts towards lower frequencies with increasing doping amount. For hexaferrites, the intrinsic resonance frequency can be described by the following equation [7]:(2)$$f_{r}=\frac{\gamma }{2\pi }\sqrt{H_{\theta }H_{\varphi }}$$
where H_θ_ and H_φ_ represent the in-plane and out-of-plane anisotropy fields of the hexaferrite, respectively. It is well known that the magnetic crystal anisotropy constants of Fe^3+^ and O^2+^ are negative, while that of Fe^2+^ is positive. With the increase of La^3+^ doping, Fe^2+^ increases to balance the charge, thereby decreasing the value, which leads to a decrease in the resonance frequency of the material.

Figure 6c depicts the tangent of the magnetic loss angle for samples with varying doping amounts. Given that the absorption mechanism of hexagonal ferrite is predicated on magnetic loss, its absorption capacity can be inferred indirectly from the variations in tan δ(μ). It can be seen that the doping of La-Sr helps to improve the absorption capacity of the material. The value of tanδ (μ) increases with the increase of the doping amount before 13 GHz, and there is a high loss at x = 0.2 and x = 0.4.

Magnetic loss generally originates from natural resonance, eddy current loss and exchange resonance for magnetic materials, and C_0_ is used to describe the magnetic loss mechanism and calculated by the following equation [20]:(3)$$C_{0}=\mu^{\prime \prime }{\left(\mu^{\prime }\right)}^{-2}f^{-1}$$
when the C_0_ value is constant at different frequencies, the magnetic loss is only contributed to by eddy current loss. From Figure 7a, it can be seen that the C_0_ curves of all samples shows a significant change in 1–4 GHz, which suggests that magnetic loss mainly comes from natural resonance. Significant fluctuations of the C_0_ value for samples with x = 0 to 0.6 in 4–16 GHz indicate that the magnetic loss of doped ferrite is mainly caused by magnetic resonance. The C_0_ value of samples with x = 0.8 and 1.0 keep constant in 4–18 GHz, indicating that eddy current loss is dominant at this frequency region.

To better assess the electromagnetic wave absorption capability of the samples, the attenuation capacity of the sample to the electromagnetic wave can be analyzed by calculating its attenuation constant according to the following formula [21]:(4)$$\alpha =\frac{\sqrt{2}\pi f}{c}\sqrt{\left(\mu^{\prime \prime }\varepsilon^{\prime \prime }-\mu^{\prime }\varepsilon^{\prime }\right)+\sqrt{{\left(\mu^{\prime \prime }\varepsilon^{\prime \prime }-\mu^{\prime }\varepsilon^{\prime }\right)}^{2}+{\left({\mu^{\prime }\varepsilon }^{\prime \prime }+\mu^{\prime \prime }\varepsilon^{\prime }\right)}^{2}}}$$
where c refers to the speed of light. The α-f graph of the sample is shown in Figure 7b. It can be seen that with the addition of La-Sr doping, the α value of the sample increases slightly in the 1–13 GHz range, but when the doping amount exceeds x = 0.6, the attenuation capacity of the sample for electromagnetic waves begins to decrease. This change is consistent with the variation of tan δ(μ). For samples with x = 0.0–0.6, as the doping increases, the grains of the sample become larger and the pores increase, which strengthens the reflection of electromagnetic waves inside the material, thus dissipating more electromagnetic waves.

Based on transmission line theory, reflection loss can be calculated using the following formula [22]:(5)$$\mathrm{RL}=20\mathrm{lg}\left|\frac{Z_{\mathrm{in}}-Z_{0}}{Z_{\mathrm{in}}+Z_{0}}\right|$$
(6)$$Z_{\mathrm{in}}=Z_{0}\sqrt{\frac{\mu_{r}}{\varepsilon_{r}}}\tanh\left(j\frac{2\pi \mathrm{fd}}{c}\sqrt{\mu_{r}\varepsilon_{r}}\right)$$
where Z_in_, Z_0_, μ_r_, ε_r_ and d refer to input impedance, free space impedance, complex magnetic conductance, complex dielectric constant and material thickness, respectively. Figure 8 shows the reflection loss of samples with x = 0.0–0.6. As shown in Figure 8b,d,f,h, it is evident that the effective absorption bands of Ba_1−x_(La_0.5_Sr_0.5_)_x_Co_2_Fe_16_O_27_ (x = 0.0, 0.2, 0.4 and 0.6, respectively) cover the C (4–8 GHz), X (8–12 GHz) and Ku (12–18 GHz) bands, and shift towards lower frequencies as the doping amount increases, which is associated with the forward shift of its inherent resonance frequency. The primary absorption band of undoped Co_2_W hexaferrite is located in the Ku band, with an RL_min_ of −29.01 dB at a thickness of 1.1 mm and an effective absorption bandwidth reaching 6.28 GHz. The primary absorption range of the sample with x = 0.2 moves forward to the X band, and the sample shows good absorption performance with the RL_min_ of −40.61 dB at a thickness of 1.5 mm and an effective absorption bandwidth of 5.76 GHz. When x = 0.4, the absorption range continues to shift towards lower frequencies to the C band, and the sample also has some absorption capability in the X and Ku bands with an RL_min_ of −37.45 dB at a thickness of 2.5 mm and a bandwidth of 4.97 GHz. Moreover, at thicknesses of 1.7 mm and 1.9 mm, the effective absorption bandwidths are 11.04 GHz and 11.93 GHz, respectively, encompassing the entire X and Ku bands, showing great potential for application. At a thickness of 2.1 mm, the sample is capable of absorbing in the C and X bands, with a bandwidth of 7.49 GHz. When x = 0.6, the RL_min_ of the sample at a thickness of 2.3 mm is −36.14, encompassing the entire X band, with an effective bandwidth reaching 5.88 GHz. Also, at thicknesses of 1.7 mm and 1.9 mm, the minimum reflection loss is around −20 dB, with good absorption capabilities in the X and Ku bands, the bandwidths being 9 GHz and 8.73 GHz, respectively. Table 2 provides a comparison of the research findings on W-type hexaferrites’ absorptive capabilities reported by other researchers in recent years. The results of the present work have certain advantages in terms of performance and tunable absorption band, which hold great potential for future applications.

## 4. Conclusions

La-Sr co-doped Ba_1−x_(La_0.5_Sr_0.5_)_x_Co_2_Fe_16_O_27_ (x = 0, 0.2, 0.4, 0.6, 0.8 and 1.0, respectively) hexaferrites were successfully prepared using the solid-state method. A series of characterizations revealed that the appropriate addition of La-Sr increases grain size and provides numerous pore defects, enabling electromagnetic waves to reflect and dissipate energy within the material. Moreover, the sample’s resonance frequency shifts towards lower frequencies with La-Sr doping, achieving absorption effects in different frequency bands and expanding the application scope. When x = 0.2, with only 1.5 mm thickness, RL_min_ is −40.61 dB, and the effective absorption bandwidth is 5.76 GHz, presenting good absorption performance in the X band. At x = 0.4, the material exhibits decent absorption performance in the C band, with a minimum reflection loss of −37.45 dB and a bandwidth of 4.97 GHz at a thickness of 2.5 mm. When x = 0.6, the material can demonstrate good absorption performance in both the X and Ku bands at a thickness of less than 2 mm. By adjusting the doping amount, absorption in different or even multiple frequency bands can be achieved, presenting significant potential for various applications.

## Figures and Tables

**Figure 1 materials-16-05897-f001:**
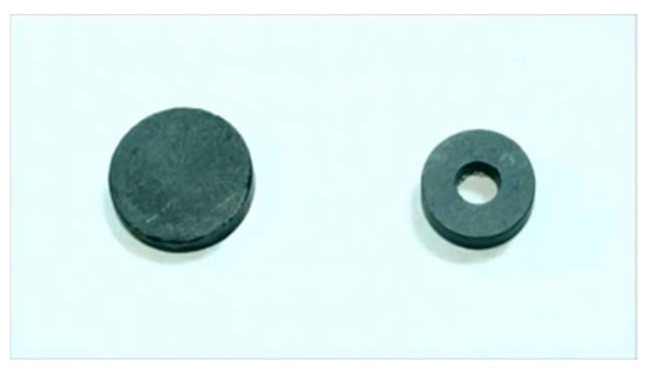
The green body of a Ba_0.6_La_0.2_Sr_0.2_Co_2_Fe_16_O_27_ hexaferrite disc (**left**) and toroidal specimen (**right**).

**Figure 2 materials-16-05897-f002:**
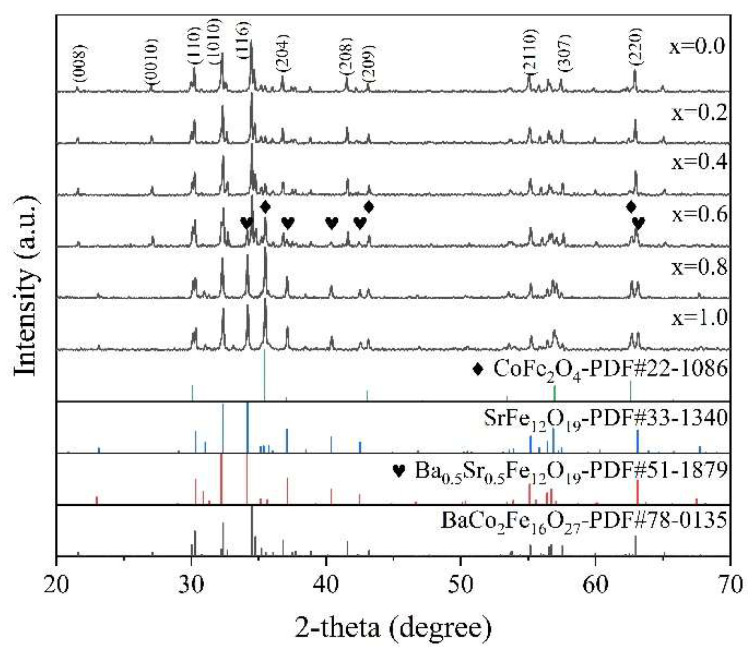
XRD patterns of hexagonal ferrite Ba_1−x_(La_0.5_Sr_0.5_)_x_Co_2_Fe_16_O_27_ (x = 0.0, 0.2, 0.4, 0.6, 0.8 and 1.0).

**Figure 3 materials-16-05897-f003:**
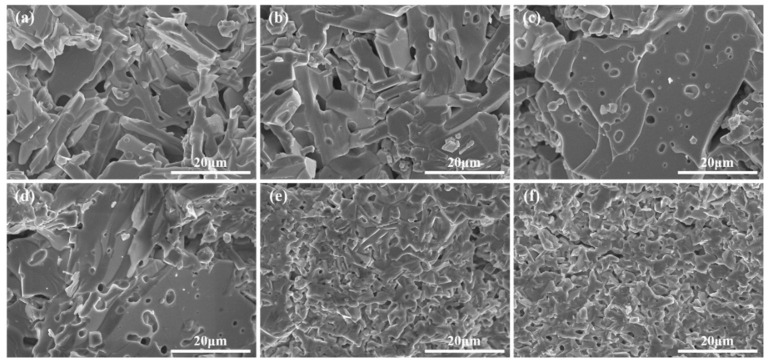
Micromorphology of Ba_1−x_(La_0.5_Sr_0.5_)_x_Co_2_Fe_16_O_27_: (**a**) x = 0.0; (**b**) x = 0.2; (**c**) x = 0.4; (**d**) x = 0.6; (**e**) x = 0.8; (**f**) x = 1.0.

**Figure 4 materials-16-05897-f004:**
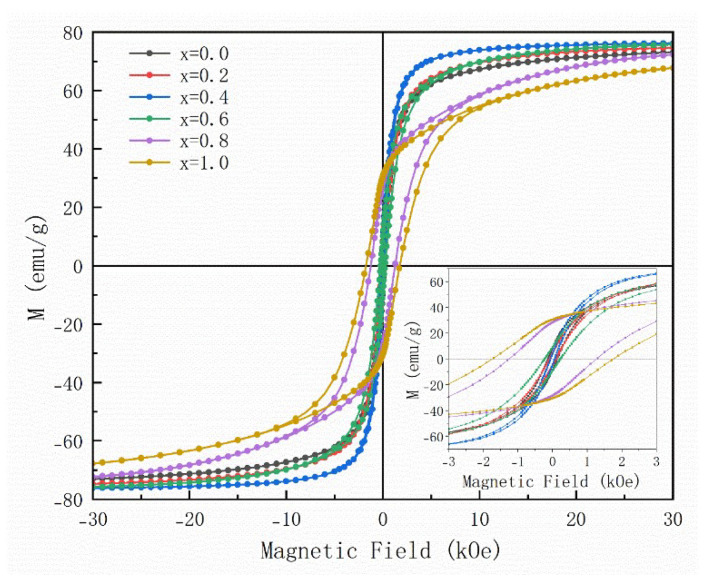
Hysteresis loops of Ba_1−x_(La_0.5_Sr_0.5_)_x_Co_2_Fe_16_O_27_ hexaferrites (x = 0.0, 0.2, 0.4, 0.6, 0.8 and 1.0, respectively) at room temperature.

**Figure 5 materials-16-05897-f005:**
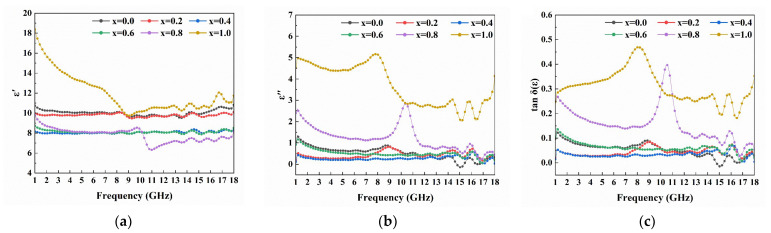
The (**a**) ε′, (**b**) ε″ and (**c**) tanδ(ε) of Ba_1−x_(La_0.5_Sr_0.5_)_x_Co_2_Fe_16_O_27_ (x = 0.0, 0.2, 0.4, 0.6, 0.8 and 1.0).

**Figure 6 materials-16-05897-f006:**
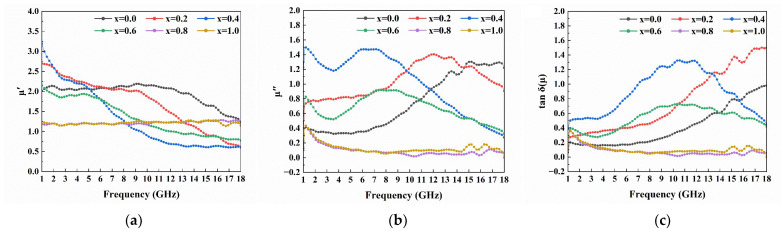
The (**a**) μ′, (**b**) μ″ and (**c**) tanδ(μ) of Ba_1−x_(La_0.5_Sr_0.5_)_x_Co_2_Fe_16_O_27_ (x = 0.0, 0.2, 0.4, 0.6, 0.8 and 1.0).

**Figure 7 materials-16-05897-f007:**
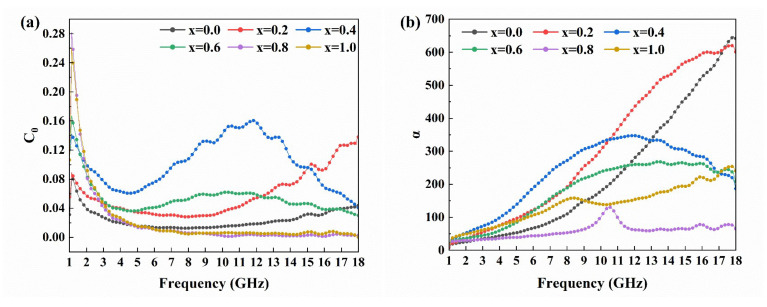
The (**a**) C_0_, (**b**) α of Ba_1−x_(La_0.5_Sr_0.5_)_x_Co_2_Fe_16_O_27_ (x = 0.0, 0.2, 0.4, 0.6, 0.8 and 1.0).

**Figure 8 materials-16-05897-f008:**
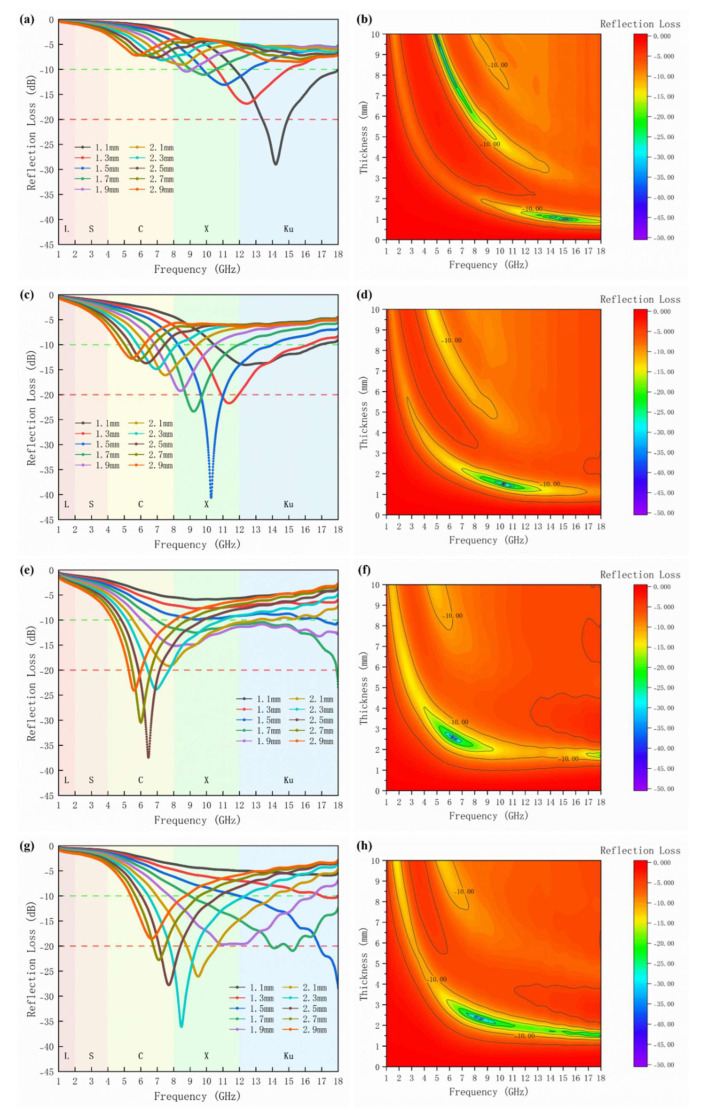
Reflection loss curves of samples with doping amount (**a**,**b**) x = 0.0; (**c**,**d**) x = 0.2; (**e**,**f**) x = 0.4; (**g**,**h**) x = 0.6 in the frequency range of 1–18 GHz.

**Table 1 materials-16-05897-t001:** Main magnetic parameters (Ms, Hc) of Ba_1−x_(La_0.5_Sr_0.5_)_x_Co_2_Fe_16_O_27_ hexaferrites (x = 0.0, 0.2, 0.4, 0.6, 0.8 and 1.0, respectively).

Sample	Ms (emu/g)	Hc (Oe)
X = 0.0	73.1	130
X = 0.2	74.7	100
X = 0.4	76.2	60
X = 0.6	75.7	230
X = 0.8	72.3	1274
X = 1.0	67.8	1752

**Table 2 materials-16-05897-t002:** Comparison of absorption performance between the results of different researchers and the present work.

Sample	Thickness (mm)	RL_min_ (dB)	Frequency (GHz)	Absorption Range (GHz)	Bandwidth (GHz)	Reference
BaZn_2_Fe_16_O_27_	3.5	−17.02	9.04	7.28–11.44	4.16	[8]
BaCo_2_Fe_15.6_(Co_0.5_Zr_0.5_)_0.4_O_27_	5.0	−37.84	16.3	14.85–17.32	2.47	[10]
BaCoMgFe_16_O_27_	4.0	−17	7.62	-	2.97	[12]
Ba_0.5_Sr_0.5_Co_1.5_Zn_0.5_Fe_16_O_27_	2.5	−35	10.4	-	-	[13]
80% Ni/20% Sr-W	2.0	−20.69	12.08	-	4	[14]
Ba_0.95_La_0.05_Co_2_Fe_15.5_Al_0.5_O_27_	7.0	−50.13	11.8	-	1.7	[17]
PANI/BaW	6.0	−40.4	2.9	-	-	[23]
Ba(MnZn)_0.4_Co_1.2_Fe_16_O_27_/NBR	2.5	−37	10.5	7.5–14	6.5	[24]
CoZnW	4.0	−35	5.8	4.3–7.7	3.4	[25]
BaZn_0.7_Co_1.3_Fe_16_O_27_	2.1	−52.8	12.9	-	4.5	[26]
ATO/BaZn_2_Fe_16_O_27_	2.8	−43.07	10.64	7.12–15.44	8.32	[27]
X = 0.0	1.1	−29.01	14.21	11.72–18	6.28	This work
X = 0.2	1.5	−40.61	10.27	8.27–14.03	5.76	This work
X = 0.4	2.5	−37.45	6.46	4.82–9.79	4.97	This work
X = 0.6	2.3	−36.14	8.48	6.47–12.35	5.88	This work

## Data Availability

The authors confirm that the data supporting the findings of this study are available within the article.

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
