# Peer review of "Absorption Band Tunable La-Sr Co-Doped BaCo2-W Type Hexaferrites"

_materials, 2023, doi:10.3390/ma16175897_

Round 1

Reviewer 1 Report

The authors have prepared La-Sr co-doped Ba1-x(La0.5Sr0.5)xCo2Fe16O27 (x = 0.0, 0.2, 0.4, 0.6, 0.8 and 1.0, respectively) hexaferrites and characterizations revealed that appropriate doping of La-Sr , absorption in different or even multiple frequency bands can be achieved, presenting significant potential for applications. The results obtained are reasonable for consideration for publication in this journal. However, there are several issues that must be clarified before the manuscript can be given further consideration for publication.

. Page-1, Line: 38:  The sentence “Compared with dielectric loss -------- of dielectric loss and magnetic loss [7]” needs to be further elaborate for the benefit of the reader to use magnetic materials.

 . Page-2,Line 57:  “Currently, most studies focus on single ion doping, while research on multi-ion co-doping is relatively limited” Explain. Is there a synthesis issue? If yes, why is that?

. Page-2,Line 85:  “For this measurement, the sintered ring-shaped specimens were machined to an outer diameter of 7 mm, an inner diameter of 3.04 mm, and a thickness of 2 mm.” It will be a good idea to show this with some pictures.

. Page-3, Line 98:  “The phase formation temperature of SrW is higher than that of BaW, so after the doping amount x>0.6, SrM becomes the main phase”. Authors should show at least one of the doped samples with detailed XRD data analysis where one can really understand the different phase formation with different ‘x’ amounts in the system.

. Page-3,Fig1: related to the above comment, figure.1 is the important figure. Authors should present at least one separate XRD data plot (x=0.4 as looks relevant to further discussion as well) with indexed peaks. There should be also a table related to the crystallographic information for all ‘x’ values from this XRD data which will give an idea about the structural properties to the readers.  

. Page-4,Line 124:  “The size of Ms mainly depends on the structure and composition of the material”. The authors should explain a bit more about this point.

Page-4,Fig3: Authors should have mentioned the temperature at which these MH loops are recorded. And what about the magnetization as a function of temperature?

Page10, Table-2: In the introduction, there are some references i.e., [12,13and 14]. Why are they excluded from this table? It will be better to include it here for comparison. If there is a reason to do that, Explain.

. All the figures should be modified as the level font size is smaller (al least for Figs. 4, 5and 6).     

Reviewer 2 Report

Presented manuscript entitled “An adsorption band tunable La-Sr co-doped BaCo2-W type hexaferrites” is written with good English and I did not find any major mistakes.

Below, please find my minor comments and suggestions:

·         Why Authors decided for La and Sr doping?

·         Lines 57-58: “Currently, most studies focus on single ion doping, while research on multi-ion co-doping is relatively limited.” – please add some references to multi-ion co-doped W-type (or any other type) hexaferrites with short description for comparison with Authors’ work.

·         Figure 1: what are the crystalline sizes for each phases?

·         Chapter 3.2.: Authors are writing about materials pores, however exact pores dimensions and volumes are not determined. I suggest performing the adsorption-desorption hysteresis loops together with specific surface area determination for obtained materials.

·         For confirmation of doping I also suggest XPS analysis, which could give the answer about atomic percentage of each element in  the material.

·         Table 1: As Authors are comparing Ms and Hc values with grains sizes, maybe adding a row with grains sizes is a good idea?

Concluding, I suggest a minor revision of presented manuscript.
